# Elongation inhibitors do not prevent the release of puromycylated nascent polypeptide chains from ribosomes

Benjamin D Hobson[1,2], Linghao Kong[1], Erik W Hartwick[3], Ruben L Gonzalez[3], Peter A Sims[1,4,5]*

[1]Department of Systems Biology, Columbia University Irving Medical Center, New York, United States; [2]Medical Scientist Training Program, Columbia University Irving Medical Center, New York, United States; [3]Department of Chemistry, Columbia University, New York, United States; [4]Department of Biochemistry & Molecular Biophysics, Columbia University Irving Medical Center, New York, United States; [5]Sulzberger Columbia Genome Center, Columbia University Irving Medical Center, New York, United States

**Abstract** Puromycin is an amino-acyl transfer RNA analog widely employed in studies of protein synthesis. Since puromycin is covalently incorporated into nascent polypeptide chains, anti-puromycin immunofluorescence enables visualization of nascent protein synthesis. A common assumption in studies of local messenger RNA translation is that the anti-puromycin staining of puromycylated nascent polypeptides in fixed cells accurately reports on their original site of translation, particularly when ribosomes are stalled with elongation inhibitors prior to puromycin treatment. However, when we attempted to implement a proximity ligation assay to detect ribosome-puromycin complexes, we found no evidence to support this assumption. We further demonstrated, using biochemical assays and live cell imaging of nascent polypeptides in mammalian cells, that puromycylated nascent polypeptides rapidly dissociate from ribosomes even in the presence of elongation inhibitors. Our results suggest that attempts to define precise subcellular translation sites using anti-puromycin immunostaining may be confounded by release of puromycylated nascent polypeptide chains prior to fixation.

*For correspondence:
pas2182@cumc.columbia.edu

Competing interests: The authors declare that no competing interests exist.

## Introduction

Subcellular localization of messenger RNA (mRNA) translation enables dynamic, rapid regulation of protein synthesis in a wide range of biological systems (*Buxbaum et al., 2015*; *Jung et al., 2014*; *Martin and Ephrussi, 2009*). The number of localized mRNAs is much greater than originally appreciated (*Lécuyer et al., 2007*), and the study of subcellular translation sites is now an area of active research across multiple organisms and cell types. A powerful, widely used technique to study protein synthesis involves the incorporation of puromycin into nascent polypeptide chains (NPC) of translating ribosomes. As an amino-acyl transfer RNA (tRNA) analog, puromycin enters the ribosomal aminoacyl-tRNA binding site (A site) and is covalently coupled to the carboxyl-activated NPC at the ribosomal peptidyl-tRNA binding site (P site) that is adjacent to the A site within the ribosomal peptidyl transferase center (PTC), terminating protein synthesis and ejecting a C-terminal puromycylated NPC (*Nathans, 1964*). This reaction was first exploited to label nascent polypeptides with fluorescent puromycin analogues (*Nemoto et al., 1999*; *Smith et al., 2005*; *Starck et al., 2004*), and became widely used both in vitro and in vivo following the development of anti-puromycin antibodies (*Schmidt et al., 2009*) and 'click'-chemistry-compatible puromycin analogues (*Liu et al., 2012*).

Despite technical advances, using puromycin labeling to visualize the subcellular distribution of nascent protein synthesis in specific cell types within complex tissues remains challenging. This is particularly true for polarized cells such as neurons, where levels of protein synthesis in dendrites and axons are relatively low compared to neighboring neuronal and glial cell bodies. Although large nerve terminals may be readily visualized (*Scarnati et al., 2018*) and expansion microscopy can provide higher resolution (*Hafner et al., 2019*), diffusion or trafficking of puromycylated NPCs prior to chemical fixation may confound attempts to determine the original translation site. To combat this problem, *David et al., 2012* introduced the ribopuromycylation (RPM) assay and used it to argue in favor of active protein synthesis within the nucleus. The premise of the RPM assay is that pretreatment of cells with elongation inhibitors, such as emetine or cycloheximide, prevents release of puromycylated nascent chains from translating ribosomes (*Bastide et al., 2018*; *David et al., 2012*). These studies assert that anti-puromycin immunostaining of labeled nascent chains in fixed cells occurs in situ on elongation-stalled ribosomes bound to mRNA.

We sought to develop a puromycin labeling assay based on RPM that would simultaneously provide cell type-specificity and preserve the ribosomal localization of the anti-puromycin signal. In the course of our studies, we found that elongation inhibitors do not prevent the release of puromycylated NPCs from ribosomes. Furthermore, structural modeling revealed that antibodies are unlikely to be able to directly contact and recognize puromycin within the PTC of intact ribosomes due to steric hindrance. Together, these results suggest that anti-puromycin immunostaining occurs only on puromycylated nascent chains released from ribosomes. Our results have important implications for the study of local translation sites using puromycin labeling.

## Results

### Implementation of a cell type-specific ribopuromycylation proximity ligation assay

To develop a cell type-specific assay for visualization of local translation sites, we envisioned combining the 'RiboTag' system (*Sanz et al., 2009*) of Cre-dependent HA-tagged Rpl22 (eukaryotic ribosomal protein L22, hereafter referred to as eL22-HA) with the RPM assay (*David et al., 2012*; *Figure 1A*). Based on the premise that elongation inhibitors prevent puromycylated NPC release from ribosomes, we reasoned puromycin would be in specific proximity to the eL22-HA antigen present on ribosomes in RiboTag-expressing cells (*Figure 1B*). Such complexes could be visualized by the proximity ligation assay (PLA), which requires two targets to be within <40 nm (*Söderberg et al., 2006*). Because the HA tag on eL22-HA is on the C-terminus, intramolecular eL22-HA/Puro PLA from nascent eL22 peptides is unlikely to contribute significant signal (*tom Dieck et al., 2015*).

We utilized murine RiboTag glioma cells (*Montgomery et al., 2019*) to test the feasibility of eL22-HA/Puro PLA in vitro. After pretreatment with emetine and brief puromycin labeling, we fixed cells and analyzed anti-puromycin immunofluorescence (IF) and eL22-HA/Puro PLA in parallel (*Figure 1C*, see *Figure 1—source data 1* for summary statistics and testing). Similar to anti-puromycin IF, the eL22-HA/Puro PLA produced a robust fluorescent signal in RiboTag glioma cells treated with puromycin, which was blocked by pretreatment with anisomycin, an antibiotic that binds to the A-site 'cleft' and competitively inhibits puromycin incorporation (*Hansen et al., 2003*; *Pestka et al., 1972*). We verified that the eL22-HA/Puro PLA signal was specific to the presence of both antibodies by omitting either the anti-puromycin or the anti-HA antibody (*Figure 1C*, see *Figure 1—source data 1* for statistics) We further tested the cell-type specificity of the eL22-HA/Puro PLA using human U87 glioma cells, which do not express eL22-HA (*Figure 1—figure supplement 1A*). As expected, these cells exhibit minimal eL22-HA/Puro PLA signal despite robust anti-puro IF labeling (*Figure 1C*). We also utilized high-resolution confocal imaging to visualize eL22-HA/Puro PLA puncta (*Figure 1D* and *Figure 1—figure supplement 2C*). Quantification of eL22-HA/Puro PLA puncta per cell revealed ~200 puncta per cell, while omission of either puromycin or HA antibodies reduced this to <8 puncta per cell (*Figure 1—figure supplement 2B*). Thus, the eL22-HA/Puro PLA accurately reports on the presence of puromycin in labeling controls (omission of puromycin/anisomycin pretreatment) and meets the standard specificity criteria of PLA (lack of signal with omission of either primary antibody or antigen).

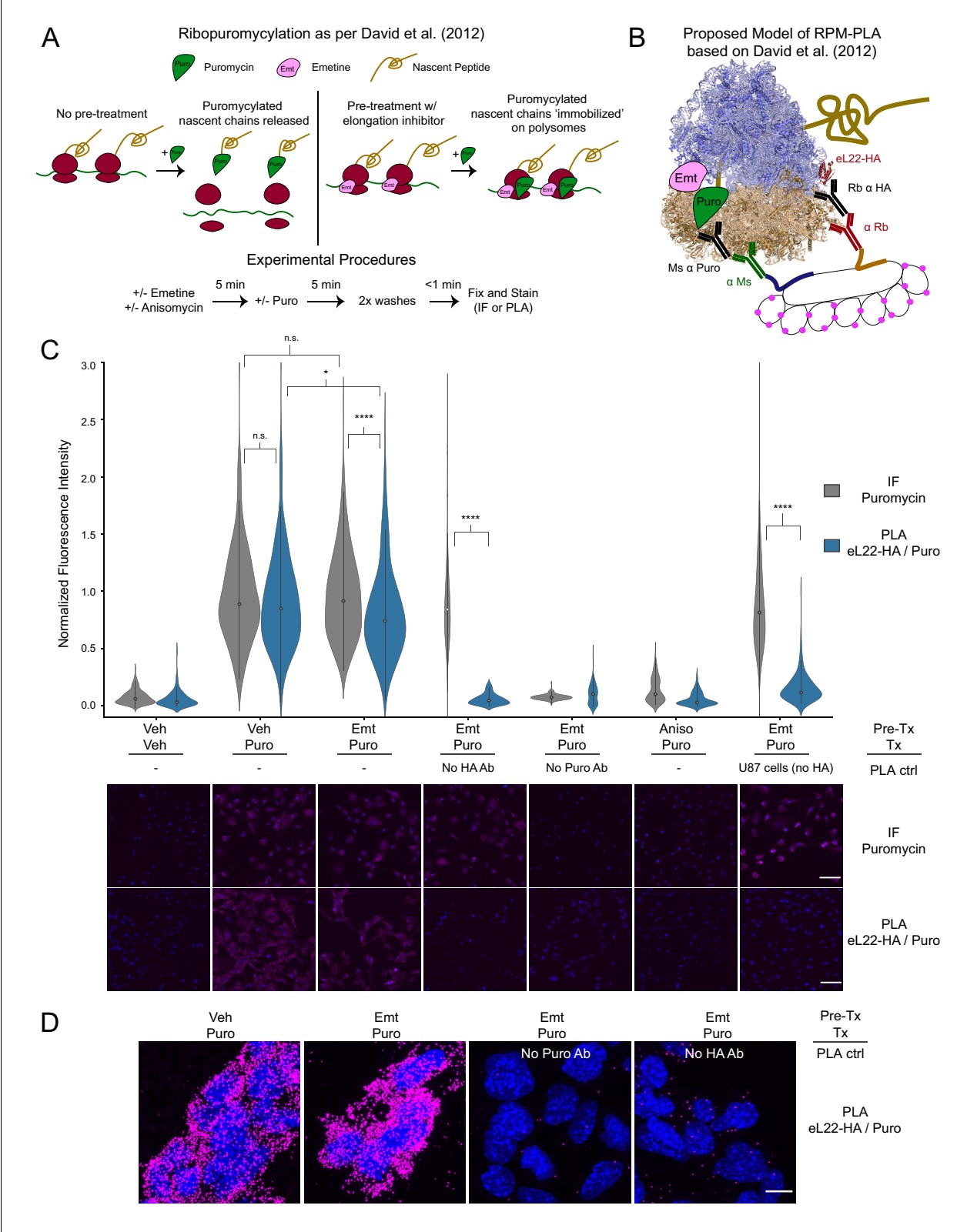

**Figure 1.** PLA targeting eL22-HA and puromycylated nascent chains produces translation- and antigen-dependent signal, but does not distinguish between emetine-stalled and untreated ribosomes. (**A**) Schematic depicting the ribopuromycylation assay as described by *David et al., 2012*. Experimental procedures used in these studies are shown below. (**B**) Schematic depicting our intended model of combining ribopuromycylation with proximity ligation assay to enable cell type-specific visualization of translation, based on *David et al., 2012*. (**C**) Violin plots of normalized fluorescence

*Figure 1 continued on next page*

*Figure 1 continued*

intensity for puromycin IF or eL22-HA/Puro PLA signal in RiboTag glioma cells treated as indicated. Exceptions are noted below the plots for omission of primary antibodies or use of U87 glioma cells (lacking eL22-HA expression). Data are derived from 3 to 4 experiments and 112–288 cells per condition, and statistical comparisons indicated were conducted using Mann-Whitney U test. See *Figure 1—source data 1* for details. * indicates p<0.05, **** indicates p<1e-4. Representative images (10x) are shown below; DAPI in blue, puromycin IF or eL22-HA/Puro PLA in magenta. Scale bar, 50 µm. (D) Representative confocal images (60x; Z-stack maximum projection) of eL22-HA/Puro PLA fluorescence for RiboTag glioma cells treated as indicated. Scale bar, 10 µm.

The online version of this article includes the following source data and figure supplement(s) for figure 1:

**Source data 1.** Statistical information pertaining to *Figure 1C* and *Figure 1—figure supplement 2C*.
**Source data 2.** Raw source data for *Figure 1C*.
**Figure supplement 1.** U87 glioma cells do not express eL22-HA, and eL22-HA/Puro PLA does not distinguish between emetine-treated and untreated cells at high resolution.
**Figure supplement 1—source data 1.** Raw source data for *Figure 1—figure supplement 1B*.
**Figure supplement 2.** Puromycin washout and Harringtonine run-off confirm that eL22HA/Puro PLA signal does not report on proximity of ribosomes and puromycylated nascent chains.
**Figure supplement 2—source data 1.** Raw source data for *Figure 1—figure supplement 2C*.

Previous imaging studies have demonstrated that in the absence of elongation inhibitors, puromycylated NPCs dissociate from ribosomes in minutes (*Morisaki et al., 2016*; *Wang et al., 2016*; *Wu et al., 2016*; *Yan et al., 2016*, see also below). Therefore, we expected that cells receiving no pretreatment would produce significantly less eL22-HA/Puro PLA signal compared to cells pretreated with emetine, which should prevent release of puromycylated nascent chains according to *David et al., 2012*. However, emetine pretreatment produced a minor, but statistically significant, *decrease* in the eL22-HA/Puro PLA signal relative to the corresponding anti-puromycin IF (*Figure 1C*) and relative to eL22-HA/Puro PLA signal from cells receiving no pretreatment (*Figure 1C*). This effect of emetine was specific to eL22-HA/Puro PLA, as emetine pretreatment did not significantly alter anti-puromycin IF (*Figure 1C*).

Surprised by these results, we conducted the eL22-HA/Puro PLA following puromycin washout (*Figure 1—figure supplement 2A*). We reasoned that after a short pulse of puromycin and an extended washout period, puromycylated NPCs would completely dissociate from ribosomes. After up to 45 min of washout, the eL22-HA/Puro PLA still produced intense signal comparable to anti-puromycin IF (*Figure 1—figure supplement 2C*). We conducted a similar experiment using harringtonine runoff (*Figure 1—figure supplement 2B*). Since the small-molecule drug harringtonine stalls newly initiating ribosomes (*Fresno et al., 1977*), extended incubation in harringtonine eliminates the presence of NPCs once elongating ribosomes are runoff. We first confirmed that harringtonine runoff removed NPCs available for puromycylation by conducting the harringtonine runoff *before* puromycin treatment (*Figure 1—figure supplement 2C*). However, conducting harringtonine runoff *after* puromycin treatment and washout still produced robust eL22-HA/Puro PLA signal that was only slightly reduced relative the corresponding anti-puromycin IF (*Figure 1—figure supplement 2C*). Therefore, we conclude that while the eL22-HA/Puro PLA is specific for the presence of antigen and antibody, it reports primarily on the cytoplasmic abundance of puromycylated NPCs, regardless of whether they are currently bound to ribosomes.

## Instability of puromycylated NPC-ribosome complexes

The results of our eL22-HA/Puro PLA were somewhat surprising, since PLA is generally assumed to produce signals based only on specific proximity (i.e. antigen-antigen interaction) given the distance requirement of <40 nm. However, recent evidence suggests that PLA signals can be generated by a variety of non-interacting antigen pairs, provided they have overlapping spatial distributions and are present at high concentrations (*Alsemarz et al., 2018*). Such a model of 'non-specific proximity' seemed to fit our PLA data, but was inconsistent with the claims of *David et al., 2012* that puromycylated NPCs remain tethered to ribosomes pretreated with elongation inhibitors. Cycloheximide binds to the ribosomal large, 60S, subunit in the tRNA 'exit' site (E site), stalling elongation by preventing exit of the deacylated E-site tRNA (*Schneider-Poetsch et al., 2010*). Emetine acts similarly to pactamycin, binding the ribosomal small, 40S, subunit in the E site and preventing translocation of the tRNA-mRNA complex (*Wong et al., 2014*). However, we are unaware of any evidence that

translocation is required for puromycin to exit the ribosome after covalently reacting with the NPC. In fact, previous imaging studies demonstrated that puromycylated NPCs are released from ribosomes in the presence of cycloheximide (*Wang et al., 2016*). To resolve the discrepancy between these findings and the claims of *David et al., 2012*, we sought biochemical evidence that elongation-inhibited ribosomes and puromycylated NPCs are stable biochemical complexes. We conducted sucrose gradient fractionation of polysomes from RiboTag glioma cells pretreated with emetine (*Figure 2A*) or cycloheximide (*Figure 2B*) prior to puromycin treatment. We found that puromycin immunoreactivity was undetectable in polysome fractions, and that puromycylated NPCs were abundant primarily in the top two fractions corresponding to free material (*Figure 2A–B*). The absence of puromycylated NPCs in polysome fractions of cells treated with puromycin, even after pretreatment with elongation inhibitors, is consistent with previous work (*Colombo et al., 1965*; *Grollman, 1968*). However, these results seem to contradict those of *David et al., 2012*, who first bound untreated polysomes to polyvinylidene fluoride (PVDF) membranes before puromycin treatment, fixation, and immuno-ELISA.

Since puromycylated NPCs are released from polysomes either prior to or during sucrose gradient centrifugation (*Figure 2A–B*), we suspected that puromycylated NPCs would also be released from polysomes bound to PVDF (*David et al., 2012*). However, puromycylated NPCs could directly bind to the PVDF matrix, creating the appearance that they are still tethered to ribosomes. We therefore conducted immunoprecipitation (IP) experiments using either anti-puromycin or anti-HA antibodies bound to magnetic beads (*Figure 2C*). RiboTag glioma cells were pretreated with emetine followed by puromycin, or pretreated with emetine, lysed, and treated with puromycin in solution. Regardless of emetine pretreatment, western blotting of captured proteins showed no evidence of co-IP: puromycin immunoreactivity was observed only in anti-puromycin IP samples, while eL22-HA and Rps6 immunoreactivity were observed only in anti-HA IP samples (*Figure 2D*). Although emetine pretreatment did not prevent release of puromycylated NPCs from ribosomes (*Figure 2A–D*), the RNA captured by eL22-HA IP confirmed that emetine inhibits polysome breakdown (*Figure 2—figure supplement 1A*). RNA captured by anti-HA IP from emetine pretreated cells had a significantly higher total yield and a lower ratio of 60S subunit 28S ribosomal RNA (rRNA) to 40S subunit 18S rRNA (28S/18S ratio), reflecting enhanced capture of intact polysomes in the presence of emetine (and enhanced capture of HA-tagged free 60S subunits in the absence of emetine). In contrast to the anti-HA IP, no detectable rRNA was captured by anti-puromycin IP, even with emetine pretreatment (*Figure 2—figure supplement 1B*). These results are consistent with the work of *Grollman, 1968* and suggest that although emetine prevents polysome breakdown by puromycin, it does not prevent release of puromycylated NPCs (*Figure 2—figure supplement 1C*).

## Direct imaging of puromycin-induced nascent chain release from ribosomes

Our biochemical studies demonstrated that puromycylated NPCs are not stably bound to ribosomes in polysome lysates, but did not reveal the kinetics of the release process or whether it occurs within puromycin-treated cells in situ. We utilized the SunTag translation reporter system (*Morisaki et al., 2016*; *Wang et al., 2016*; *Wu et al., 2016*; *Yan et al., 2016*) in HEK-293FT cells in order to directly visualize puromycylated NPC release (*Figure 3A*). We employed the version of this system described by *Wu et al., 2016*, which comprises (i) an mRNA encoding the SunTag array of 24 GCN4 repeats followed by oxBFP and an auxin-induced degron (AID), (ii) super folder green fluorescent protein (sfGFP)-bound single-chain variable fragment (scFv) of GCN4 antibody (scFv-GCN4-sfGFP), and (iii) Oryza Sativa transport inhibitor response 1 (OsTIR1) to remove completed proteins containing the AID (*Figure 3A*). Because the soluble scFV-GCN4-sfGFP binds rapidly to the SunTag array (*Tanenbaum et al., 2014*) present in NPCs as they emerge from the ribosome, polysomes translating the reporter are visualized as bright spots (hereafter SunTag puncta) on top of the cytoplasmic background fluorescence (*Figure 3B*). Consistent with previous work (*Morisaki et al., 2016*; *Wang et al., 2016*; *Wu et al., 2016*; *Yan et al., 2016*), we found that the SunTag puncta disappear in minutes upon treatment with puromycin (*Figure 3B*). Thus, the fading of SunTag puncta into the cytoplasmic background is caused by GFP-bound puromycylated NPCs diffusing away from polysome complexes (*Figure 3A*).

To study the kinetics of puromycylated NPC release, we conducted time lapse imaging in live cells, continuously cycling between 7–12 cells per dish. We pretreated cells with emetine or

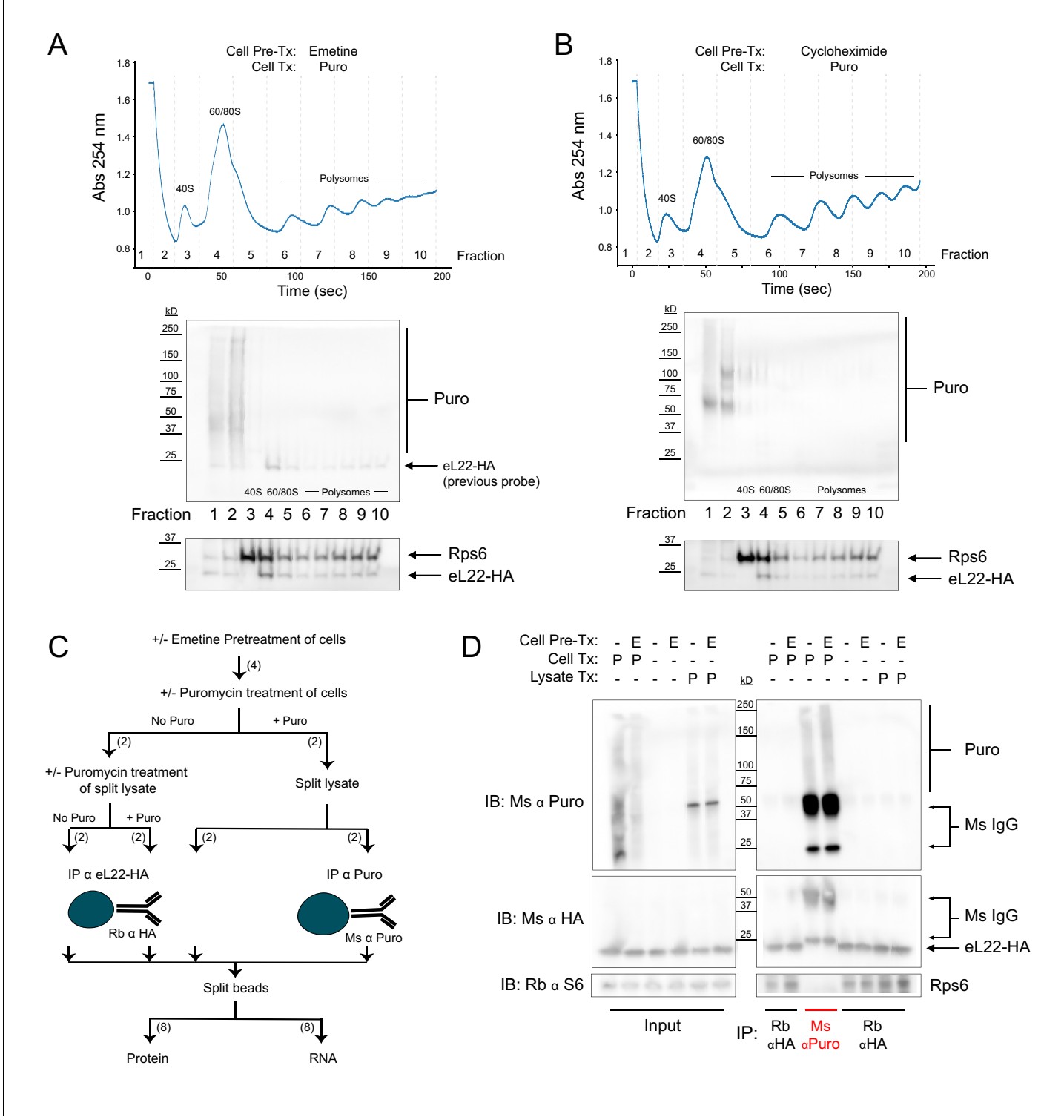

**Figure 2.** Regardless of pretreatment with elongation inhibitors, ribosomes and puromycylated nascent chains are not stable biochemical complexes. (**A**) Sucrose gradient fractionation of polysomes from cells pretreated with emetine prior to puromycin. *Top*: Absorbance spectra from 15–50% linear sucrose gradient. Dashed lines indicate fractions collected, which correspond to western blots below. *Bottom*: Western blots for puromycin, Rps6, and eL22-HA. Note that Rb α S6 and Ms α HA were probed first, and some residual Ms anti-HA remained on the membrane during subsequent probing with Ms α Puromycin (see identical staining pattern below). (**B**) Same as (**A**), but with cycloheximide pretreatment. (**C**) Schematic depicting experimental workflow for eL22-HA and puromycin co-immunoprecipitation experiment. Numbers in parentheses indicate the number of samples along each branch of the workflow. All indicated splits of lysates and beads were equal, such that each RNA or protein sample represents an equal input of cell lysate. (**D**)
*Figure 2 continued on next page*

*Figure 2 continued*

Western blots of puromycin, eL22-HA, and Rps6 from input and IP samples as indicated. In blots using Ms α Puromycin or Ms α HA, strong bands corresponding to Ms IgG are present in the Ms α Puro lanes. Note that the Ms IgG light chain (~25 kD) is clearly resolved from eL22-HA (~23 kD, *Sanz et al., 2009*).

The online version of this article includes the following source data and figure supplement(s) for figure 2:

**Source data 1.** Raw western blot images pertaining to *Figures 2A, B and C*.

**Figure supplement 1.** Pretreatment with emetine increases yield of intact ribosomes in eL22-HA immunoprecipitation, but does not enable capture of ribosomal RNA in puromycin immunoprecipitation.

**Figure supplement 1—source data 1.** Raw source data for *Figure 2—figure supplement 1A*.

cycloheximide prior to the onset of imaging, and then added puromycin 60 s into each 8 min imaging trial. We observed complete loss of SunTag puncta in all puromycin-treated cells within this timeframe (*Figure 3C*), with minimal loss of SunTag puncta in untreated control cells (*Figure 3C*) and negligible photobleaching in all conditions (*Figure 3—figure supplement 1B*). Representative single-cell trajectories are shown in *Figure 3—figure supplement 2*. Replicate imaging trials for all drug treatment conditions were highly reproducible (*Figure 3—figure supplement 1C*), enabling kinetic analysis of puromycylated NPC release. Consistent with previous work (*Wang et al., 2016*), we found that cycloheximide pretreatment slowed, but did not prevent, the disappearance of SunTag puncta (*Figure 3C*). We note that our assay cannot distinguish between the rate of puromycin binding to nascent chains and the rate of their release from ribosomes. A previous report on cycloheximide-pretreated polysome lysates found that 50% of nascent peptides were released in a 30 s exposure to puromycin, while complete release was observed with a 5 min exposure (*Colombo et al., 1965*). The same 30 s puromycin exposure caused complete release of nascent chains from untreated polysomes (*Colombo et al., 1965*), suggesting that cycloheximide slows the rate of puromycin binding to ribosomes and/or to nascent chains. These observations are consistent with our results (*Figure 3C*) and with recent work by *Wang et al., 2016*, who proposed that the conformation of the ribosomal pre-translocation complex that is stabilized by cycloheximide causes crowding of the A site and slows the ability of puromycin to covalently react with the NPC. Although we cannot exclude that cycloheximide also slows the release kinetics of NPCs once they have been transferred to puromycin, we found no evidence that elongation is required for the release process (*Figure 3C*).

In contrast to cycloheximide, emetine pretreatment enhanced the initial puromycin-induced release kinetics compared to puromycin alone (*Figure 3C*). Although emetine pretreatment eliminated the apparent lag phase seen in cells treated with puromycin or cycloheximide-puromycin, it slightly slowed the overall decay rate compared to puromycin alone. Emetine pretreated cells lost one-third of SunTag puncta significantly faster than those treated with puromycin alone (*Figure 3D*, Welch's t-test for time at 66.6% remaining: $t(5) = 3.932$, p=0.0271), but the effect was smaller for half-life (Welch's t-test for time at 50% remaining: $t(5) = 3.183$, p=0.0294) and was no longer significant when two-thirds of puncta were lost (Welch's t-test for time at 33.3% remaining: $t(5) = 0.967$, p=0.3996). Although the mechanisms underlying these kinetic differences remain unknown, it is possible that emetine-bound ribosomes have either a higher affinity for puromycin and/or an enhanced reaction efficiency upon binding. Such a model is consistent with previous work describing enhanced puromycin incorporation in the presence of emetine (*David et al., 2012*), particularly at low temperatures (*David et al., 2013*). Regardless, these data clearly show that puromycylated NPCs are rapidly released from ribosomes even when cells are pretreated with emetine.

## Anti-puromycin IgG molecules are unlikely to directly contact and recognize puromycin within the PTC of intact 80S ribosomal complexes

*David et al., 2012* did not propose a model for where the puromycin is physically located during fixation and immunostaining in the RPM procedure. For puromycylated NPCs to truly remain 'tethered' to ribosomes, the puromycin would have to be localized in either the PTC or in the proximity of the nascent polypeptide exit tunnel. Despite the rapid kinetics of puromycylated NPC release, it is conceivable that a few puromycin-terminated NPCs might become chemically fixed inside ribosomes. We therefore conducted structural modeling to assess whether it is feasible that an anti-puromycin

IgG molecule could directly contact and recognize puromycin within an intact 80S ribosomal complex. It is inconceivable that the antigen binding fragment (Fab) of an immunoglobulin G (IgG) molecule could access puromycin within the narrow, constricted geometry of the exit tunnel, so we focused our structural modeling on the PTC (*Figure 4A*). We used a 2.8 Å cryogenic electron microscopy (cryo-EM) structure of a stalled mammalian 80S ribosomal complex containing a P-site tRNA (*Chandrasekaran et al., 2019*) and a 2.8 Å X-ray crystallographic structure of an intact mouse IgG2a monoclonal antibody (*Harris et al., 1997*), the exact isotype of the anti-puromycin clone 12D10 used in our study and in many others (*Schmidt et al., 2009*).

After we positioned puromycin within the PTC of the 80S ribosomal complex (see Materials and methods), we docked the IgG2a Fab within the A site such that the antigen binding site of the variable fragment was within ~6 Å or closer of puromycin. Such a model immediately presented with obvious steric clashes between the Fab and multiple sites encompassing the 60S subunit ribosomal proteins, 28S rRNA, and P-site tRNA (*Figure 4C–D*). We analyzed the steric clashes resulting from twelve distinct Fab positions within the A site (see Materials and methods) using MolProbity to analyze ribosomal and tRNA residues located within a 12 Å radius of the Fab (*Chen et al., 2010*; *Williams et al., 2018*). The MolProbity Clashscores, the number of serious steric overlaps (>0.4 Å) per 1000 atoms, ranged from 85.79 to 246.3 (*Figure 4B*). All of these MolProbity Clashscores were in the $0^{th}$ percentile of all structures solved by cryo-EM to any resolution in the Protein Data Bank, that is there were no cryo-EM structures solved to any resolution with worse Clashscores (*Figure 4B*, *Figure 4—source data 1*). In contrast, the individual structures of the 80S ribosomal complex and IgG2a each have MolProbity Clashscores < 21 (*Figure 4B*) and are in the $87^{th}$ percentile of all cryo-EM structures solved to any resolution or of all X-ray crystallographic structures solved to similar resolutions, respectively (*Figure 4—source data 1*). Next, we attempted to model the Fab fragment so it was positioned as close as possible to puromycin within the A site of the 80S ribosomal complex but in such a manner so as to minimize steric clashes. This model generated a MolProbity score of 22.89, representing the $25^{th}$ percentile of all cryo-EM structures solved to any resolution (*Figure 4B*, *Figure 4—source data 1*). Most strikingly, for this 'best' fit scenario, the Fab is positioned 29.1 Å away from puromycin, well outside the distance that would be required for the antigen binding site of the Fab to directly contact puromycin via non-covalent interactions. Thus, the numerous steric clashes between the Fab and the A site of the ribosome would preclude positioning of the antigen recognition site close enough to recognize puromycin within the PTC of intact ribosomal complexes.

It is worth noting that our modeling was conducted using a stalled 80S ribosomal complex in which the ribosomal subunits are in their 'non-rotated' conformational state and the P-site tRNA is in its 'classical' configuration (*Chandrasekaran et al., 2019*). Although the non-rotated/classical conformation is likely favored in ribosomal complexes stalled by cycloheximide (*Budkevich et al., 2011*) and emetine (*Jiménez et al., 1977*), it is conceivable that a minor sub-population of ribosomal complexes becomes chemically fixed in the so-called 'rotated' conformational state in which the tRNAs occupy their 'hybrid' configurations. In this case, the relative rotation of the subunits dramatically narrows the aperture of the A site, making it even less likely that the IgG Fab could access the PTC. Overall, these results suggest that anti-puromycin antibodies are unlikely to access the interior of intact ribosomes during immunostaining.

## Discussion

Although we successfully visualized puromycin labeling in RiboTag glioma cells, the most appealing applications of the eL22-HA/Puro PLA were based on the premise that each translating ribosome could be visualized with its NPC in situ. Unfortunately, we found no evidence to support the RPM model proposed by *David et al., 2012*. Our results have important implications for the study of subcellular translation sites using puromycin labeling, since the RPM model is often cited as evidence that puromycylated NPCs cannot move away from translation sites in the presence of elongation inhibitors (e.g. *Moissoglu et al., 2019*). The first major problem with the RPM model is the kinetics of puromycylated NPC release. We found that the half-life of puromycylated NPC-ribosome complexes is less than 40 s when cells are pretreated with emetine, and extended by only ~2 min with cycloheximide (*Figure 3D*). Although the precise mechanisms underlying these kinetic differences are unknown, our results strongly suggest that once the carboxyl-activated NPC is coupled with puromycin, no further translocation or dissociation of the 80S complex is required for puromycylated

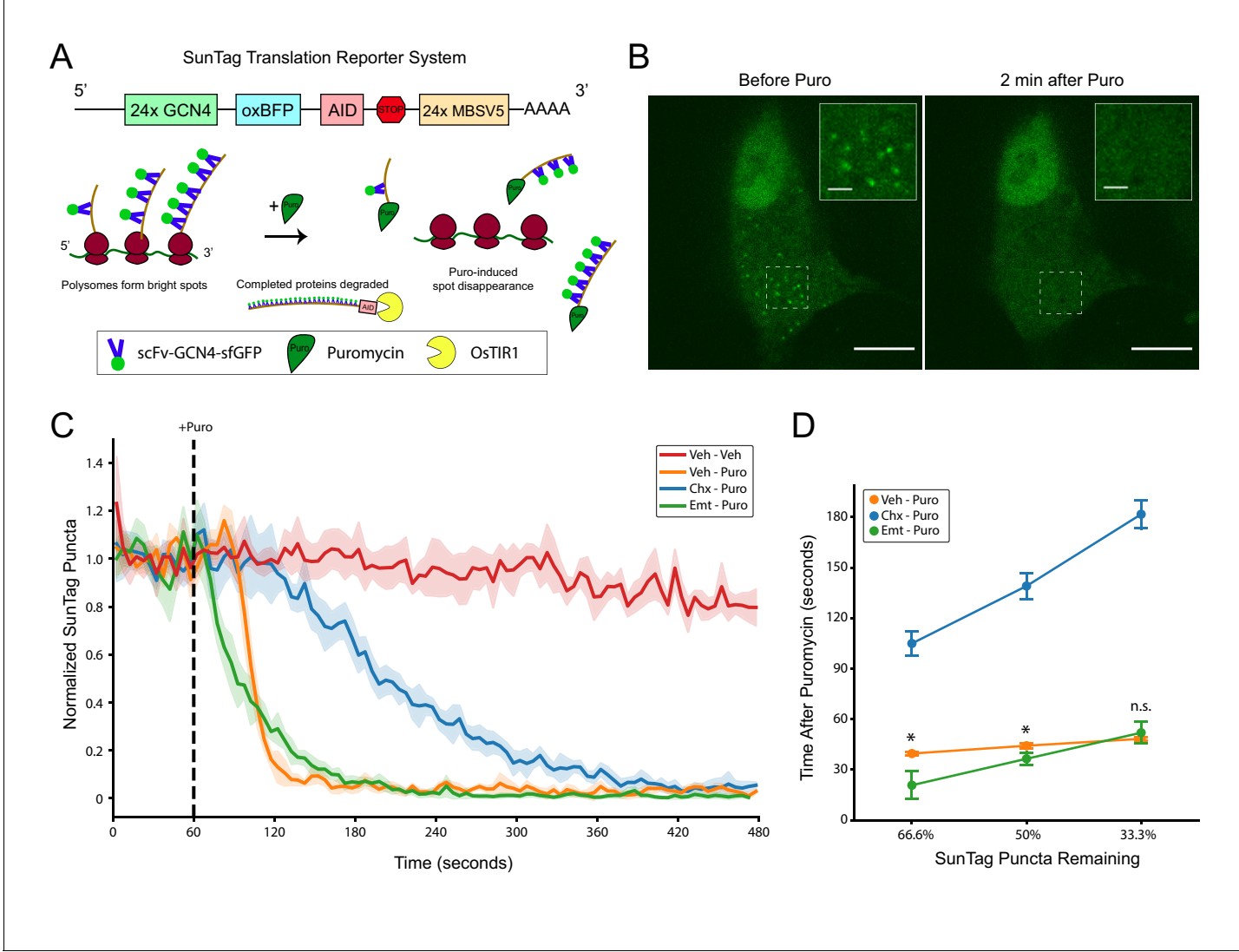

**Figure 3.** SunTag translation reporter imaging shows that elongation inhibitor pretreatment does not prevent puromyclated nascent chain release from polysomes. (A) Schematic depicting the SunTag translation reporter system used in this study, as described by *Wu et al., 2016*. Note that although the SunTag reporter construct harbors MS Coat Protein Binding Sites (MBSV5) in the 3' UTR, we did not employ RFP-labeled MS Coat Protein for mRNA tracking. (B) Representative spinning disk confocal images (100x) of a live HEK-293FT cell before and 2 min after addition of 220 μM puromycin. Regions depicted in white dashed lines are displayed in the inset. Scale bars: 10 μm for large field, 2 μm for inset. (C) Live imaging time course of SunTag puncta in cells with pretreatments as indicated (355 μM cycloheximide or 54 μM emetine), followed by treatment as indicated (220 μM puromycin). Puromycin was added at 60 s into the 8 min imaging trial (dashed black line). SunTag puncta for each cell were normalized to the average SunTag puncta in that cell during the initial 60 s. Data are comprised of 3–4 replicate imaging trials per treatment condition, with each replicate containing 7–12 cells (Veh-Veh: n = 27 cells from three replicates, Veh-Puro: n = 30 cells from three replicates, Chx-Puro: n = 27 cells from three replicates, Emt-Puro: n = 41 cells from four replicates). Plotted are the mean ± standard deviation of all cells, computed in five second time intervals (see Materials and methods). (D) The mean normalized SunTag puncta in each replicate imaging trial was used to determine the time after puromycin addition at which two thirds, half, and one third of SunTag puncta remain in each condition. Welch's t-test was used to assess differences between Veh-Puro and Emt-Puro (for 66.6%: $t(5) = 3.932$, $p=0.0271$, for 50%: $t(5) = 3.183$, $p=0.0294$, for 33.3%: $t(5) = 0.967$, $p=0.3996$). * denotes $p<0.05$. The online version of this article includes the following video, source data, and figure supplement(s) for figure 3:

**Source data 1.** Raw source data for *Figure 3C and D*.

**Figure supplement 1.** SunTag puncta detection, minimal photobleaching, and consistency of replicate imaging trials.

**Figure supplement 2.** Representative images and traces of single cell SunTag imaging.

**Figure 3—video 1.** Time lapse live imaging of puromycin-induced SunTag puncta disappearance.

https://elifesciences.org/articles/60048#fig3video1

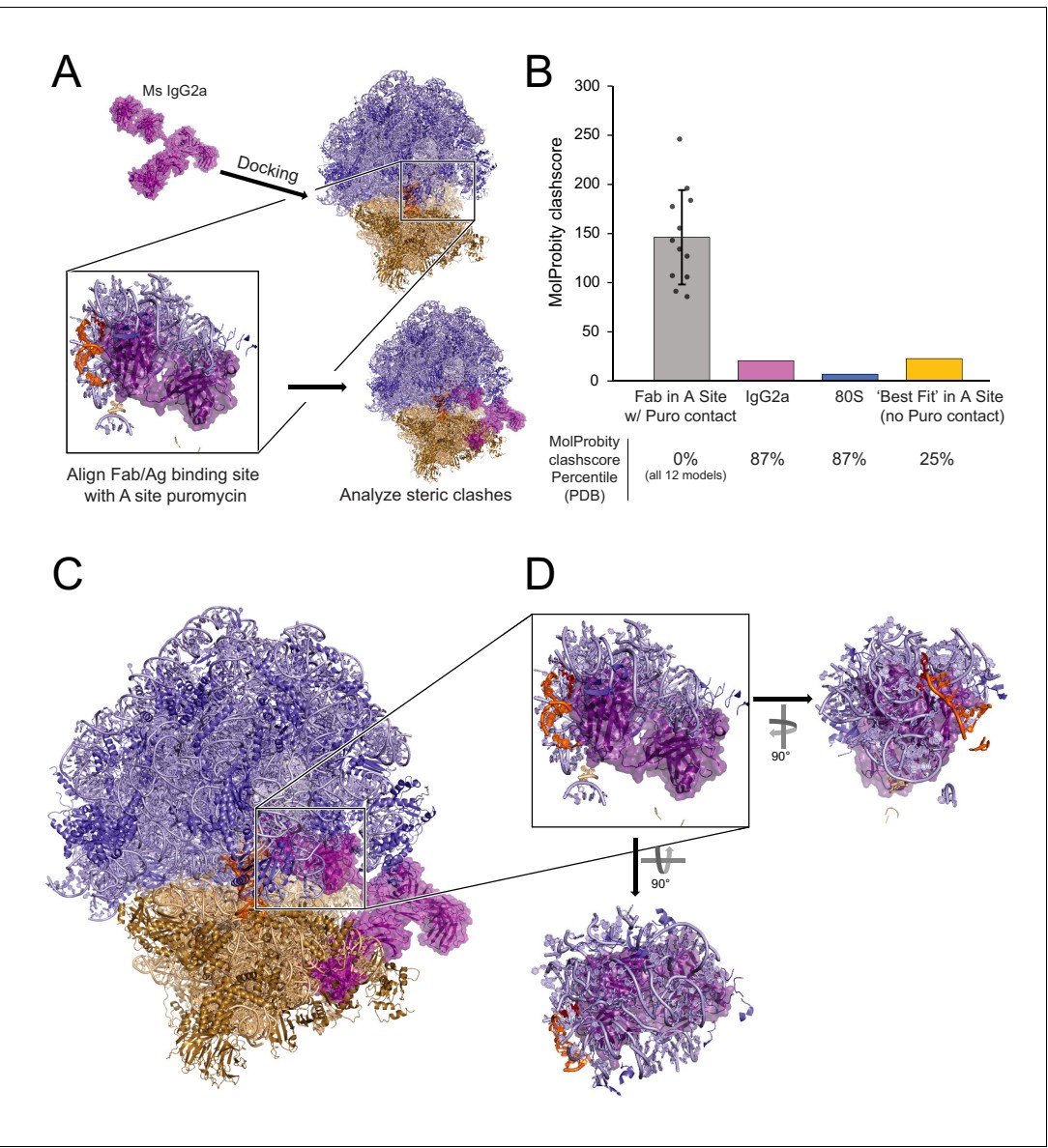

**Figure 4.** Structural analysis does not support a model of IgG antibodies directly contacting and recognizing puromycin within intact 80S ribosomal complexes. (**A**) Schematic depicting structural modeling workflow: puromycin was positioned within the PTC of a stalled mammalian 80S ribosomal complex and the Fab fragment of a mouse IgG2a antibody was docked into the A site such that the antigen recognition loop could directly contact puromycin (see Materials and methods). Twelve such positions were analyzed by MolProbity. (**B**) MolProbity clashscores, the number of serious steric overlaps (>0.4 Å) per 1000 atoms, for the A Site Fab positions in which the Fab could contact puromycin (within ~6 Å), the IgG2a alone, the 80S alone, or the 'best fit' model of the Fab in the A site in which the Fab cannot contact puromycin (29.1 Å away). Mean ± standard deviation are plotted for the 12 models with puromycin contact. MolProbity clashscore percentiles (the percent of structures of comparable resolution in PDB with worse clashscores) are listed below. See *Figure 4—source data 1* for detailed summary. (**C**) Representative positioning of IgG2a antibody manually docked in the A site of the 80S ribosomal complex. (**D**) One of the twelve representative models of the IgG2a Fab within the A site of the 80S ribosomal complex highlighting the numerous steric clashes with the 60S ribosomal proteins, 28S rRNA, and P-site tRNA.

The online version of this article includes the following source data for figure 4:

**Source data 1.** Summary of MolProbity analysis of IgG2a Fab alignments within 80S A site.

NPC release (*Figure 2*, *Figure 2—figure supplement 1*). We suspect that puromycin is small enough to simply bypass the P site tRNA and proceed directly from the PTC through the exit tunnel. Critically, all SunTag puncta were undetectable at ~5 min after puromycin in all conditions (*Figure 3C*). Even with short incubations at lower puromycin concentrations, labeled NPCs would continue to be released during washing and fixation. Furthermore, complete cytoplasmic fixation can take longer than 15 min at room temperature when using conventional 4% paraformaldehyde to fix monolayer cultures (*Huebinger et al., 2018*). Thus, it is likely that the vast majority of puromycylated NPCs are not crosslinked to their ribosomal origin during chemical fixation. The second major problem with the RPM model is that antibodies are unlikely to directly contact and recognize puromycin within the PTC of fixed, intact ribosomes (*Figure 4*). Thus, even if a small sub-population of puromycin molecules are retained in the PTC after chemical fixation, our structural modeling suggests that these puromycin molecules would be inaccessible during immunostaining.

Based on the kinetics of puromycylated NPC dissociation (*Figure 3*) and inaccessibility of intra-ribosomal puromycin to antibody binding (*Figure 4*), there should be no general expectation that anti-puromycin immunostaining accurately reports on the original ribosomal translation site. However, our results do not preclude the possibility that released puromycylated NPCs are bound by neighboring proteins, organelles, and/or ribonucleoprotein complexes immediately adjacent to their translation site. Such nascent polypeptide binding complexes would include canonical targeting motifs such as the signal recognition particle, cytoplasmic chaperones (e.g. *Hansen et al., 1994*), and neuron-specific RNA granules (e.g. *Graber et al., 2013*). Indeed, puromycin-resistant SunTag puncta have been observed in dendrites, and these puncta are thought to contain stalled polyribosomes and NPCs (*Langille et al., 2019*). Given that emetine does not impede puromycylated NPC release (*Figure 3*), it is not surprising that the labeling of these complexes by anti-puromycin IF did not require emetine (*Langille et al., 2019*). Instead, the persistence of puromycin at these sites must require context-specific binding of NPCs within these ribonucleoprotein complexes. Since anti-puromycin IF is unlikely to occur within the ribosome (*Figure 4*), it is likely that these nascent polypeptide interactions occur outside of the ribosome.

Given the implausibility of a true 'RPM-PLA' as we had envisioned (*Figure 1*), the results of our eL22-HA/Puro PLA also contribute to growing concerns that PLA can produce false positive signals when both antigens are highly abundant and have overlapping subcellular distributions (*Alsemarz et al., 2018*). This issue of 'non-specific proximity' may broadly affect puromycin-based PLAs (e.g. *tom Dieck et al., 2015*), as was the case in our study. Ribosomes are the most abundant macromolecular machine in the cell, with typical cultured cells estimated to harbor at least 1 million ribosomes (*Wada et al., 2011*). Accordingly, even a brief treatment with puromycin will produce an immense number of puromycylated NPCs per cell. Viewed through this lens, it is not surprising that our eL22-HA/Puro PLA could produce ~200 puncta per cell based solely on non-specific proximity (*Figure 1—figure supplement 1B*). However, it should be noted that our assay is an outlier in this regard, as both antigens are extremely abundant and broadly distributed throughout the cytoplasm. The use of direct-conjugated primary antibody PLA probes would be expected to reduce the frequency of false positives. We remain optimistic that development of new technologies for imaging subcellular translation sites in specific cell-types will accelerate studies of local translation in complex tissues.

## Materials and methods

### Cell culture

The murine RiboTag glioma line expressing eL22-HA was a gift from the laboratory of Dr. Peter Canoll and has been previously described (*Montgomery et al., 2019*). Briefly, gliomas were generated by injection of PDGFA-IRES-Cre expressing retrovirus into subcortical white matter of mice harboring floxed p53, RiboTag, and stop-flox mCherry-luciferase alleles. Glioma cells isolated from the retrovirus-induced tumor of a male mouse were expanded in culture to establish the murine RiboTag glioma line. HEK-293FT cells were obtained from ThermoFisher (Cat# 5700–07, RRID:CVCL_6911). U-87MG (U87) human glioma cells (ATCC cat# HTB-14, RRID:CVCL_0022) were a gift from the laboratory of Dr. Ramon Parsons. U87 cells, murine RiboTag glioma cells, and HEK-293FT cells were

cultured in Dulbecco's modified eagle medium (DMEM, Life Technologies, catalog #11965118) supplemented with 10% fetal bovine serum (FBS, Life Technologies, catalog #16000044).

Plasmids and Drugs pUbC-FLAG-24xSuntagV4-oxEBFP-AID-baUTR1-24xMS2V5-Wpre was a gift from Robert Singer (Addgene plasmid # 84561; RRID:Addgene_84561). pUbC-OsTIR1-myc-IRES-scFv-sfGFP was a gift from Robert Singer (Addgene plasmid # 84563; RRID:Addgene_84563). The following drugs were obtained from Sigma: Anisomycin (catalog #A9789), Cycloheximide (catalog #C7698), Emetine dihydrochloride (catalog number #E2375), and Puromycin (catalog #P7255). Harringtonine was obtained from Santa Cruz Biotechnology (catalog #sc-204771).

## Cell treatment with translation inhibitors

Cells were seeded 24–48 hr prior to treatment and were approximately 70–80% confluent at the time of experiments. Pretreatments were conducted for 5 min as described in the most recent RPM methods paper (*Bastide et al., 2018*). Cycloheximide was used at 355 µM, and anisomycin at 37 µM. Emetine was used at 45 µM, as suggested by *Bastide et al., 2018*, which is ~4.6 times lower than the 208 µM used in *David et al., 2012*. Note that maximal inhibition of translation is achieved at 1 µM emetine (*Grollman, 1968*), and 45 µM emetine is thus a saturating concentration for inhibiting elongation. Pretreatment inhibitors were maintained at the same concentration throughout puromycin treatment and washing.

After pretreatment, cells were then treated with 10 µM puromycin for 5 min. Both pretreatment and treatment occurred at 37°C. Following the pretreatment and treatment, cells were immediately and quickly washed twice in ice-cold 1X phosphate buffered saline (PBS) with 5 mM $MgCl_2$, then fixed in 1X PBS with 5 mM $MgCl_2$, 4% sucrose, and 4% paraformaldehyde (PFA) for 15 min at room temperature. The samples were then washed three times with 1X tris-buffered saline (TBS). After washing, cells were used for immunofluorescence staining or proximity ligation assay as described below.

For the puromycin washout experiments, cells were treated with 10 µM puromycin for 5 min, then washed twice with fresh media, then incubated in fresh media for 15 or 45 min. For the harringtonine runoff experiments, cells were treated with 10 µM puromycin for 5 min, then washed twice with fresh media, then incubated in fresh media with 100 µM harringtonine for 20 min. Cells that did not receive puromycin prior to harringtonine were treated with 10 µM puromycin for 5 min after 20 min of harringtonine treatment. Treatment with puromycin and harringtonine occurred at 37°C. Cells were then washed and fixed as described above.

## Immunofluorescence

After fixation samples were incubated in blocking/permeabilization buffer consisting of 1X TBS with 10% normal goat serum (NGS) and 0.1% Tween 20. Following blocking, samples were incubated with primary antibodies in 1x TBS with 2% NGS and 0.1% Tween 20. Primary antibodies were used as follows: monoclonal mouse anti-puromycin (clone 12D10, MilliporeSigma, catalog #MABE343, RRID:AB_2566826) at 1:500, human anti-ribosomal P (Immunovision, catalog #HPO-0100) at 1:3000, and rabbit anti-HA (Abcam, catalog #ab9110, RRID:AB_307019) at 1:500 – for 60 min at room temperature. Samples were then washed three times with 1X TBS. Next, samples were incubated with secondary antibodies in 1X TBS with 2% NGS and 0.1% Tween 20 for 30 min at room temperature. Secondary antibodies were used as follows: goat anti-mouse Alexa647 (Invitrogen, catalog #A-21236, RRID:AB_141725) at 1:1000, goat anti-rabbit A488 (Invitrogen, catalog #A-11034, RRID:AB_2576217) at 1:1000, and donkey anti-human Cy3 (Jackson ImmunoResearch, catalog #709-165-149, RRID:AB_2340535) at 1:500. Following secondary antibody incubation, cells were washed three times with 1X TBS at room temperature, then stored in DAPI Fluoromount G (Southern Biotech, catalog #0100–20) for <24 hr at 4°C prior to imaging.

## Proximity ligation assay

PLA (Duolink, MilliporeSigma) was performed in accordance with the manufacturer's instructions with slight modifications to accommodate simultaneous immunofluorescence. Blocking and permeabilization were first conducted as described for immunofluorescence above. Further blocking was conducted using Duolink blocking buffer at 37°C for 30 min. Primary antibodies and concentrations were used as in immunofluorescence above (Rabbit anti-HA, Mouse anti-Puromycin, Human anti-

Ribosomal P), but diluted in Duolink Antibody Diluent. PLA probe incubation, ligation, amplification, and final washes were conducted as per manufacturer's instructions. Anti-Rabbit MINUS and Anti-Mouse PLUS probes (MilliporeSigma, catalog #DUO92004 and #DUO92001, respectively) were used in conjunction with Duolink In Situ Detection Reagents Far Red (MilliporeSigma, catalog #DUO92013). Following the final washes in Duolink Wash Buffer B, samples were incubated in 1X TBS with 2% NGS and 0.1% Tween 20 plus donkey anti-human Cy3 at 1:500 for 20 min at room temperature. Finally, the cells were washed three times with 1X TBS and stored in Duolink Mounting Media with DAPI (MilliporeSigma, catalog #DUO82040) for <24 hr at 4°C prior to imaging.

## Immunofluorescence/PLA image acquisition and quantification

Widefield imaging of anti-puromycin IF or eL22-HA/Puro PLA was conducted on a Nikon Ti2 Eclipse equipped with a SpectraX light engine (Lumencor) and a DS-Qi2 camera (Nikon), using a 10x/0.25 NA or 20x/0.75 NA air objectives (Nikon) as indicated in the figure legends. Confocal imaging was conducted on a Leica SP8 laser scanning system using a 60x/1.45 NA objective (Leica).

Each treatment condition had 3–5 replicates, and at least 32 individual cells per replicate were selected randomly across a minimum of three fields. Cells were manually segmented using the anti-Ribo P IF profile, and the average puromycin signal intensity (from either anti-puromycin IF or eL22-HA/Puro PLA) was acquired from each cell. For each experiment, the highest average anti-puromycin IF (or eL22-HA/Puro PLA) intensity across all conditions was used to normalize each individual cell's anti-puromycin IF (or eL22-HA/Puro PLA) intensity. Note that this normalization was conducted separately for anti-puromycin IF and eL22-HA/Puro PLA signal intensities. The highest average anti-Puromycin (or eL22-HA/Puro PLA) intensity was always from cells treated with puromycin alone or from cells treated with emetine + puromycin. The normalized signal intensities from 3 to 5 separate experiments are reported. See *Figure 1—source data 1* for complete summary of replicates, cell numbers, and statistical information.

## Cell treatment and lysis for biochemical studies

Cells were pretreated with cycloheximide or emetine as described above for immunofluorescence/PLA. Cells were then treated with 100 µM puromycin for five minutes, at which point media was removed and cells were lysed in ice cold polysome buffer (150 mM KCl, 10 mM MgCl$_2$, 5 mM HEPES pH 7.4, 1% Igepal CA-630) supplemented with EDTA-free protease inhibitor cocktail (Roche, catalog #5056489001) and SUPERaseIN (Invitrogen, catalog #AM2696). For cells receiving elongation inhibitor pretreatments, cycloheximide and emetine concentration was maintained (355 µM and 45 µM, respectively) in polysome lysis buffers/sucrose gradients throughout the entire procedure. After rotating polysome lysates for 15 min at 4°C, lysates were clarified by centrifugation at 16,000xg for 15 min at 4°C. For the polysome immunoprecipitation experiment, the clarified polysome lysates from cells not receiving puromycin were split in half. Half the lysate was incubated with 100 µM puromycin for five minutes at room temperature and then returned to ice. Clarified polysome lysates were then subjected to polysome immunoprecipitation or sucrose gradient fractionation as described below.

## Sucrose gradient fractionation

Clarified polysome lysates were loaded onto 15–50% sucrose gradient and centrifuged at 37,000 RPM in a SW41 rotor for 3.5 hr at 4°C. Polysome gradients were fractionated and the optical density at 254 nm was continuously recorded using Isco-UA5 fluorescence/absorbance monitoring system. Ten fractions were collected from top to bottom of the gradient with approximately 1 mL volume each. Samples were concentrated to approximately 0.3 mL using a speed-vac, mixed with 0.1 mL of 4x Laemmli Sample Buffer, boiled for 5 min at 95°C, and stored at −80°C until western blotting.

## Polysome and puromycin immunoprecipitation

Dynabeads Protein G (Invitrogen, catalog #10004D) were washed two times in polysome buffer. Protein G beads were then incubated with an excess of primary antibodies (Rabbit anti-HA or Mouse anti-Puromycin) in polysome buffer for 20 min at room temperature. Unbound antibodies were removed by washing the beads three times in polysome buffer, and antibody-bound beads were then added to clarified polysome lysates (equivalent amounts of beads/antibodies were used for

each IP). Antigen capture was conducted for one hour at 4°C, after which the supernatant was discarded. Beads were further washed four times at 4°C in polysome buffer supplemented with protease and RNAse inhibitors (as above for cell lysis).

After the final wash, beads were split into equal aliquots for RNA and protein analysis. For RNA analysis, RNA was eluted from the beads with RLT buffer and purified using RNEasy MinElute columns (Qiagen, catalog #74204). RNA was eluted into 10 µL of nuclease free water and analyzed on Agilent 2100 Bioanalyzer using the RNA 6000 Pico kit (Agilent, catalog #5067–1513). For protein analysis, protein was eluted from the beads by boiling for 5 min at 95°C in Laemmli Sample Buffer (BioRad, catalog #161–0747). Proteins were stored at −80°C until western blotting.

## Western blotting

Equal volumes of polysome fractions or bead eluates were separated on 4–15% polyacrylamide gradient gels (BioRad, catalog #4561086) and transferred to PVDF membranes (Immobilon-P, Millipore-Sigma, catalog #IPVH00010). Membranes were initially washed for 15 min in TBST (1X TBS + 0.1% Tween 20), blocked for an hour in 5% BSA/TBST, and incubated overnight at 4°C with primary antibodies in 5% bovine serum albumin/TBST overnight. Primary antibodies were used as follows: Mouse anti-HA (Cell Signaling, catalog #2367S) at 1:1000, Mouse anti-Puromycin (clone 12D10, Millipore-Sigma, catalog #MABE343, RRID:AB_2566826) at 1:1000, and Rabbit anti-S6 Ribosomal Protein (clone 5G10, Cell Signaling, catalog #2217, RRID:AB_331355) at 1:1000. After primary incubation, membranes were washed three times in TBST prior to incubation with horseradish peroxidase (HRP)-conjugated secondary antibodies in 5% non-fat dried milk/TBST for one hour at room temperature. Secondary antibodies were used as follows: Goat Anti-Mouse HRP (Jackson ImmunoResearch, catalog #115-035-003, RRID:AB_10015289) at 1:5000, and Goat Anti-Rabbit HRP (Jackson ImmunoResearch, catalog #111-005-003, RRID:AB_2337913) at 1:5000. After secondary incubation, membranes were washed three times in TBST. Signal was developed using Immobilon enhanced chemiluminescent substrate (Millipore, catalog #WBKLS0500) and imaged on an Azure Biosystems C600 system. Where membranes were re-probed with different primary antibodies, 0.05% sodium azide with primary antibodies overnight to quench residual HRP bound to membranes.

## SunTag cell transfection, drug treatment, and image acquisition

HEK-293FT cells were transfected approximately 48 hr before imaging using CalFectin (SignaGen Laboratories, catalog number #SL100478). A plasmid ratio of 3:1 for the SunTag:scFv-sfGFP/OsTIR1 plasmids was used, while maintaining the total DNA concentration and reagent ratios recommended by the manufacturer. Transfection media was replaced with fresh media after 14 hr. Cells were split 12 hr after the media change and $0.6 \times 10^6$ cells were seeded into 35 mm glass dishes for imaging (MatTek, catalog #P35G-1.5–10 C). Indole-3-acetic acid (IAA; Sigma, catalog #I5148) was added to the medium at 500 µM overnight.

The following morning immediately prior to imaging, cells were briefly washed once with pre-warmed Leibowitz's L15 media (ThermoFisher, catalog #11415064). Cells were then placed in warm L15 media supplemented with 500 µM IAA and transferred to a pre-warmed, humidified stage-top incubator maintained at 37°C (Tokai Hit). Imaging was conducted using a 100X/1.45 NA oil-immersion objective on a Nikon TiE Eclipse equipped with a spinning disk confocal unit (Yokogawa CSU-X1) and an EM-CCD camera (Photometrics Evolve 512).

Between six and twelve fields per dish were selected for continuous imaging, which resulted in an image of each field on average every $5.4 \pm 1.9$ s (mean ± standard deviation, range 2.5–9.8 s). Pre-treatment with cycloheximide (355 µM) or emetine (54 µM) was initiated 4 min prior to the onset of image acquisition. Puromycin treatment (220 µM) was initiated 1 min after the onset of image acquisition (drug pretreatment was therefore 5 min). The total length of image acquisition was approximately 8 min.

## SunTag image processing and quantification

Time lapse image stacks were quantified for the presence of cytoplasmic SunTag puncta using the freely available ImageJ plugin TrackMate v4.0.1 (*Tinevez et al., 2017*). We used the Laplacian of Gaussian spot detector with estimated blob diameter of 0.5 µm and initial quality threshold of 300. Additional filtering was implemented using a combination of quality, contrast, and total intensity as

necessary to suppress nuclear spot detection. *Figure 3—figure supplement 1B* shows representative SunTag puncta detection. The average cytoplasmic GFP intensity was quantified across the time lapse images. Cells showing greater than 10% decrease in cytoplasmic GFP intensity, either due to photobleaching or loss of focal plane, were discarded from subsequent analysis (nine such cells were discarded out of 140 total cells imaged). Representative single cell traces and images are shown in *Figure 3—figure supplement 2*.

Exported spot statistics and cytoplasmic GFP intensity for each frame were aligned to the appropriate timestamps from the image metadata. SunTag puncta counts for all frames from each cell were normalized to the average of SunTag puncta counts of that cell during the first 60 s (pre-puromycin addition). Cytoplasmic GFP intensity was normalized to the average of the first 10 frames. The normalized SunTag puncta counts from each experimental replicate (consisting of 7–12 cells) were smoothed using a boxcar filter (rolling window mean) with a window width of k = 10 observations. Specific decay times ($t_{2/3}$, $t_{1/2}$, $t_{1/3}$) were calculated for each replicate by averaging the time stamps of all frames within ±0.01 of the relevant smoothed normalized SunTag puncta values (i.e. for $t_{1/2}$, 0.49–0.51). Comparisons between decay times of drug treatments were made using an independent samples Welch's t-test. For plotting, five-second interval bins were used to compute the mean and standard deviation across drug treatment replicates.

To create a single movie featuring multiple fields with distinct, non-uniform frame rates (*Figure 3—video 1*), the frames from each time lapse image stack were duplicated to the lowest common denominator so that all stacks had the same frame number. The time of each frame acquisition in each image stack was stamped on the images, along with the time of puromycin addition, before merging the image stacks and down-sampling to 102 frames. The resulting stack was converted to an AVI file with JPEG compression at eight frames per second, resulting in a time lapse rate of approximately 40x (~8 min of imaging played in ~12 s).

## Structural modeling

Modeling of the mouse IgG2a puromycin antibody (*Harris et al., 1997*) (PDB ID: 1igt) bound to a mammalian 80S ribosomal complex containing a peptidyl-transfer RNA (tRNA) in the ribosomal peptidyl-tRNA (P) site (*Chandrasekaran et al., 2019*) (PDB ID: 6sgc) and carrying puromycin in the ribosomal aminoacyl-tRNA (A) site (*Hansen et al., 2002*) (PDB ID: 1q82) was performed using PyMOL (*Burley et al., 2019*; *Schrödinger, 2015*). Briefly, PDB 1q82 was aligned to PDB 6sgc using the large ribosomal subunit RNA (rRNA) to position puromycin within the A-site side of the ribosomal peptidyl transferase center (PTC). We then defined a smaller region of the IgG2a antibody, referred to as the Fab (*Sela-Culang et al., 2013*), that consisted of two variable domains from the light and heavy chains ($V_H$ and $V_L$ respectively) and two constant domains ($C_H1$ and CL, respectively) corresponding to residues 1–214 and 1–230 from chains A and B, respectively. 12 models of the Fab-, puromycin-, and P-site tRNA-bound ribosomal complex were then manually generated such that the position of the Fab within the A site was varied in each model while maintaining the Fab complementarity determining regions (CDRs) within ~6 Å or closer to the puromycin moiety. These conservative constraints would enable a direct interaction, but prevent strong van der Waals clashes, between the CDRs and puromycin. A 12 Å radius centered on the Fab was then analyzed using MolProbity (*Williams et al., 2018*), an all-atom structure validation tool, in order to assess all-atom clashes by calculating the MolProbity clashscore and percent of structures of comparable resolution with worse MolProbity clashscores. The calculated clashscores, percent of structures of comparable resolution with worse MolProbity clashscores, and clash sites were used for the data reported in *Figure 4—source data 1* and the structural models and bar graph reported in *Figure 4*.

## Statistical analysis

Comparisons of immunofluorescence and PLA intensity distributions in were conducted using the Mann-Whitney U-test. Comparison of rRNA yields and 28 s/18 s ratios was conducted using Welch's independent samples t-test. Comparison of SunTag puncta decay times was conducted using Welch's independent samples t-test. Details regarding replicates, cell numbers, and test statistics are provided in the figure captions (except see *Figure 1—source data 1* for summary of testing presented in *Figure 1C* and *Figure 1—figure supplement 2C*). Statistical significance was considered at p<0.05.

## Acknowledgements

We would like to thank Theresa Swayne and Laura Munteanu for expert technical assistance with SunTag imaging, which was conducted in the Confocal and Specialized Microscopy Shared Resource of the Herbert Irving Comprehensive Cancer Center at Columbia University, supported by NIH grant #P30 CA013696. LK received support from the Columbia University II Rabi Scholars Program. This work was supported by NIH grants F30 DA047775-02 (BDH) and R33CA202827 (PAS).

## Additional information

### Funding

| Funder | Grant reference number | Author |
| --- | --- | --- |
| National Institutes of Health | R33CA202827 | Peter A Sims |
| National Institutes of Health | F30 DA047775 | Benjamin D Hobson |

The funders had no role in study design, data collection and interpretation, or the decision to submit the work for publication.

### Author contributions

Benjamin D Hobson, Conceptualization, Formal analysis, Supervision, Validation, Investigation, Methodology, Writing - original draft, Writing - review and editing; Linghao Kong, Formal analysis, Investigation, Methodology, Writing - review and editing; Erik W Hartwick, Ruben L Gonzalez, Data curation, Formal analysis, Investigation, Methodology, Writing - review and editing; Peter A Sims, Conceptualization, Resources, Supervision, Funding acquisition, Project administration, Writing - review and editing

### Author ORCIDs

Benjamin D Hobson https://orcid.org/0000-0002-2745-5318
Ruben L Gonzalez https://orcid.org/0000-0002-1344-5581
Peter A Sims https://orcid.org/0000-0002-3921-4837

### Decision letter and Author response

Decision letter https://doi.org/10.7554/eLife.60048.sa1

## Additional files

### Supplementary files

• Transparent reporting form

### Data availability

All data generated or analyzed during this study are included in the manuscript and supporting files. Source data files have been provided for all Figures.

The following previously published datasets were used:

| Author(s) | Year | Dataset title | Dataset URL | Database and Identifier |
| --- | --- | --- | --- | --- |
| Harris LJ, Larson SB, Hasel KW, McPherson A | 1997 | Structure of Immunoglobulin | https://www.rcsb.org/structure/1igt | RCSB Protein Data Bank, 1IGT |
| Hansen JL, Schmeing TM, Moore PB, Steitz TA | 2003 | Crystal Structure of CC-Puromycin bound to the A-site of the 50S ribosomal subunit | https://www.rcsb.org/structure/1Q82 | RCSB Protein Data Bank, 1Q82 |
| Chandrasekaran V, Juszkiewicz S, Choi J, Puglisi JD, | 2019 | Rabbit 80S ribosome stalled on a poly(A) tail | https://www.rcsb.org/structure/6SGC | RCSB Protein Data Bank, 6SGC |

Brown A, Shao S,
Ramakrishnan V,
Hegde RS

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
