## [Decision Letter]

**Acceptance summary:**

The manuscript details the reasons and mechanism why detection of puromycin cannot be used as a tracer for translational activity, even in the presence of elongation inhibitors.

**Decision letter after peer review:**

Congratulations, we are pleased to inform you that your article, "Elongation Inhibitors do not Prevent the Release of Puromycylated Nascent Polypeptide Chains from Ribosomes", has been accepted for publication in *eLife*.

The manuscript details the reasons and mechanism why detection of puromycin cannot be used as a tracer for translational activity, even in the presence of elongation inhibitors.

Reviewer #1:

The authors of this, and the co-submitted manuscript investigated the assumption that puromycin is a spatial indicator of translating ribosomes. This assumption is based on the puromycylation of nascent chains remaining associated with the ribosome on the mRNA, in effect freezing in its position there. When detected by antibodies to the puromycin derivative, the signal supposedly indicated the sites of translation. In this particular manuscript, the authors attempted to use this in combination with a proximity ligation assay to determine the components at the stalled translation site. Being unable to reproduce the expected result of biotinylated translation components, but only finding labeled nascent chains they reviewed their assumptions and discovered the source of the discrepancy.

The work presented here convincingly shows that the signal comes from released nascent chains, truncated by the puromycylation. For instance, in a staightforward experiment using the suntag system, the presence of labeled nascent chains did not include the signal when the tagged puromycin was used, but rather only after they were released from the ribosome indicating that they immediately terminated upon incorporating the puromycin, even in the presence of elongation inhibitors. They further modeled the ability of an antibody to access the puromycylated peptide in the ribosome and found it would not have been detectable given the steric clash.

The work is of significance particularly since a number of high impact publications have relied on the signal indicating the location of ongoing translation. Conclusions that derive from that assumption have to be re-evaluated, since some of them are clearly wrong (eg translation in the nucleus) and other, particularly in neurons, misleading. I feel it is important to publish this work before more erroneous conclusions contaminate the literature.

Reviewer #2:

In this study the authors show convincingly that translation elongation inhibitors do not prevent the release of puromycylated nascent chains. These findings impact the assay to assess the intracellular site of translation based on the localization of puromycin conjugated polypeptides, as previous studies had suggested that such puromycin conjugated polypeptides are still bound to the ribosome on which they were made, and thus represent the site of translation.

Overall, this study presents a very comprehensive and convincing set of data that show that, in fact, that localization of puromycin-conjugated polypeptides generally does not represent the site of synthesis, but rather, these polypeptides have released from the ribosome and have diffused away from their site of synthesis. I agree with the authors that this is an important point to make that will benefit the field.

My only issue with this study is that the main point, that translation inhibitors do not prevent release of polypeptides from the ribosome in the presence of translation inhibitors, has already been demonstrated convincingly previously, for example in Wang et al., 2016, which is also cited by the authors. The current studies certainly goes beyond this (and other) previous studies; first, the current study presents many additional types of analysis confirming this point, including elegant structural analysis of the ribosom. But perhaps more importantly, previous studies (e.g. Wang et al., 2016) have not specifically emphasized the relevance of these findings for assays using puromycin localization as a proxy for the site of translation.

Reviewer #3:

Hobson et al. provide strong evidence to question conclusions on local translation from experiments done with the ribopuromycylation assay (RPM). Using biochemistry and single-molecule imaging approaches , they demonstrate that translation elongation inhibitors (emetine and cycloheximide) do not prevent the release of puromycylated nascent peptides from ribosomes, which diffuse away from the translating ribosome. Hence, localization of puromycylated peptides in cells using anti-puromycin antibodies do not correspond to the site where they are synthesized. Structural modeling of the ribosome indicates steric impediments for puromycin antibodies to recognize the puromycylated nascent chain in the ribosome. Additionally, the authors point that proximity ligation assays (PLA) of highly abundant proteins might report on their overlapping distribution instead of their specific interaction.

Conclusions from this work are further supported by a submitted manuscript from Rachel Green's group. Both articles warn caution to conclude on localized translation from RPM assays. Studying localized translation in cells and organisms is a growing field that requires tools able to faithfully report, at the subcellular level, the site of translation. Therefore, their findings should be shared with the scientific community, so previous conclusions can be revisited, and test with biochemistry and imaging approaches detailed in these manuscripts.

The biochemical, single-molecule imaging and structural modeling experiments are well done, analyzed and described. It will be nice to see the localization of the mRNA in Figure 3.